# DEEP LEARNING AIDED BROADCAST CODES WITH FEEDBACK

## ABSTRACT

Deep learning aided codes have been shown to improve code performance in feedback codes in high noise regimes due to the ability to leverage non-linearity in code design. In the additive white Gaussian broadcast channel (AWGN-BC), the addition of feedback may allow the capacity region to extend far beyond the capacity region of the channel without feedback, enabling higher data rates. On the other hand, there are limited deep-learning aided implementations of broadcast codes. In this work, we extend two classes of deep-learning assisted feedback codes to the AWGN-BC channel; the first being an RNN-based architecture and the second being a lightweight MLP-based architecture. Both codes are trained using a global model, and then they are trained using a more realistic vertical federated learning based framework. We first show that in most cases, using an AWGN-BC code outperforms a linear-based concatenated scheme. Second, we show in some regimes, the lightweight architecture far exceeds the RNN-based code, but in especially unreliable conditions, the RNN-based code dominates. The results show the promise of deep-learning aided broadcast codes in unreliable channels, and future research directions are discussed.

## 1 INTRODUCTION

Error correcting codes are an integral component of wireless communication systems, allowing reliable communication over noisy channels. Codes should be designed to be as efficient as possible without sacrificing error correction capabilities. In this work, we focus on designing codes for the additive white Gaussian noise broadcast channel (AWGN-BC) with feedback. Unlike the single user feedback case, where the capacity of the AWGN channel with addition of feedback cannot exceed that of the AWGN channel without feedback, the use of feedback in the AWGN-BC channel can far exceed the capacity of the AWGN-BC channel without feedback Shannon (1956); Ahmad et al. (2015). The design of good codes for the AWGN-BC channel with feedback can enable high data rate communication with high reliability in unfavorable channel conditions.

Codes designed for the AWGN-BC channel with feedback have been linear in nature due to more simplicity in code design, yet this requirement limits the design of optimal codes Ahmad et al. (2015). These codes include Ozarow's extension of Shakwijk and Kailath's linear scheme to the AWGN-BC channel with perfect feedback Ozarow & Leung-Yan-Cheong (1984); Schalkwijk & Kailath (1966). There are also linear control-oriented schemes that show performance improvement beyond Ozarow's scheme Elia (2004); Ardestanizadeh et al. (2012). Most purely linear schemes operate under the assumption of perfect feedback and suffer greatly with feedback noise. In response to this, a concatenated scheme relying on linear codes for the inner code for the AWGN-BC for the noisy feedback channel was proposed Ahmad et al. (2015).

Dropping the linearity assumption of these codes could unlock the higher data rates of AWGN-BC codes with potentially noisy feedback, but designing non-linear codes is a challenging task. Recent innovations in deep learning, though, allow seemingly intractable wireless problems to be solved and can be used to design high performing non-linear codes. For example, in the single user AWGN channel with feedback, it has been shown that deep-learning aided codes can outperform linear schemes in terms of probability of block error in scenarios where feedback noise is high and/or the forward SNR is low Kim et al. (2020). In this work, we seek to design codes that outperform linear schemes in the aforementioned noisy regions using deep-learning for the AWGN-BC channel, where

there are limited implementations of deep-learning aided broadcast codes, in order to improve per user data rate and reliability in poor channel conditions.

## 1.1 Existing Deep-Learning Aided Broadcast Codes and Feedback Codes

There are very few existing implementations of deep-learning aided codes for the AWGN-BC channel. In Li *et. al*, a deep-learning based approach to the fading AWGN broadcast channel with two users and binary phase shift keying (BPSK) modulation is proposed. The scheme involves two phases, where in the first phase two separate time slots are allocated for transmission to each of two users, where the information bits are modulated using BPSK and the weights of the modulated bits are optimized. In phase 2, a deep learning aided two-phase scheme is used which involves passing the input through a deep recurrent neural network (RNN), through a dense layer, and then through a weighted parameter. The decoding scheme involves two bi-directional GRUs, followed by a bi-drectional LTSM, and finally a dense layer Li et al. (2022). This scheme is not fully learnable and restrictive as it requires an initial BPSK modulation scheme. For the multiple-access channel (MAC), Ozfatura *et. al* propose a deep-learning based code with two users with noiseless feedback Ozfatura et al. (2023). In this work, block attention feedback (BAF) codes are used, where the encoding and decoding process consists of transforming the knowledge vector at the receiver and transmitters into a sequence of vectors and using self-attention mechanism on these to emulate encoding and decoding processes. BAF codes have high computational complexity which may not be suitable for many communication applications, and in many applications feedback codes could benefit from complexity reduction Ozfatura et al. (2022).

On the other hand, there are numerous schemes for deep-learning codes for the single user AWGN channel with feedback. One scheme, *Deepcode*, was proposed for a finite length, fixed rate block code over the AWGN channel with noisy feedback Kim et al. (2020). In this scheme, the encoder and decoder are both modeled as recurrent neural networks (RNNs) in order to process the bit stream sequentially over the AWGN channel to minimize block error rates, but it appears to be limited to rate $1/3$ only. In Kim et al. (2023), a RNN-based power constrained deep-learning architecture is proposed. This work combines GRUs and an attention mechanism to optimize code design across time and take advantage of noise averaging. In addition, a *power control layer* is proposed, to ensure optimal utilization of the power budget. In Ozfatura *et al.*, generalized block attention feedback codes (GBAF) were proposed, which is a transformer-based feedback code scheme Ozfatura et al. (2022). GBAF operates generally by breaking message blocks into symbols, encoding individual symbols with feature extractors, and then encoding these symbols across the entire codeblock using a transformer based architecture. Transformer-based and RNN-based encoders and decoders, in general, may have large computational demands, which may not be feasible for communication systems with limited resources. To this end, LightCode was proposed, whose performance is shown to be comparable or, in some cases, superior to GBAF with fewer parameters and complexity Ankireddy et al. (2024). Unlike the aforementioned codes which utilize RNN or transformer based architectures, LightCode is a relatively lightweight symbol by symbol scheme consisting of a feature extractor (FE) architecture followed by an multi-layer perceptron (MLP) at the encoder and decoder.

## 1.2 Implementation of Deep-Learning Aided AWGN-BC Codes with Feedback

Now, we discuss our implementation of codes for the AWGN-BC channel. As mentioned before, the existing implementation in Li et al. (2022) is not fully learnable, imposing restrictive structure to the learned scheme, so we look to the existing single user AWGN with feedback codes in our design. In this work, we focus on extending the Robust Power-Constrained Deep Learning Algorithm (hereby referred to as the RPC scheme) in Kim et al. (2020) and LightCode in Ankireddy et al. (2024) to the AWGN-BC channel. We choose to evaluate the performance of RPC due to its robustness to noise as it employs a GRU and attention mechanisms to leverage noise averaging which may be useful in adverse channel conditions Kim et al. (2020). Conversely, since the model in RPC may contain many hidden states requiring a great deal of memory, we choose to compare the performance against the newly proposed LightCode as it has demonstrated near state of the art performance in certain regimes and at a code rate of $R = 1/3$ while using a less memory intensive architecture Ankireddy et al. (2024).

In the case of both codes, we demonstrate an extension of these codes to two receivers by implementing multiple decoder modules and adjusting the loss function accordingly. In this way, the entire system consisting of the encoder and decoders is treated as a global model in which the purpose of training is to minimize the overall loss between all users. We train each model for two users, and show the performance of each algorithm in different noise regimes and for various code rates.

Beyond this, we view vertical federated learning (VFL) as a promising direction for training feedback codes across the AWGN-BC channel, instead of utilizing a global model framework as outlined above. To the best of our knowledge, there have been no deep learning aided feedback codes which have employed a federated training approach. A federated framework could allow the implementation of deep learned feedback codes in practice– for example, it could allow the training of codes when decoders do not know the number of other users in the system *a priori*. Also, a federated training approach could offload computation from end-user devices which may be more resource constrained to the base station. This could also allow adaptive learning of codes in dynamic wireless environment Niknam et al. (2020). We consider the encoder and decoders each to have their own local model of the communication system and view the encoder as the *active* party. In this case, all parties share the same sample space, but each party's local model is updated according to it's own loss function Liu et al. (2024). To this end, we propose a VFL training framework where each decoder transmits its model output to the encoder over the feedback link and, conversely, the encoder transmits the new computed gradients to each decoder at the end of a batch. In the proposed VFL algorithm, we transmit the uncoded gradients across the downlink channel which are corrupted by AWGN. We choose to explore the uncoded downlink versus a quantized method as it has been shown that there is better convergence behavior in federated learning with noisy downlinks Amiri et al. (2021). We observe the effect of noise in the VFL training process.

In summary, we discuss the implementation and training of the RPC-based and the LightCode-based AWGN-BC code both for the global model and the extension to the VFL framework. Results are compared to the concatenated scheme in Ahmad *et al.* since it is designed for noisy feedback specifically, whereas purely linear codes suffer greatly with feedback noise Ahmad et al. (2015). We compare the performance of each scheme in various SNR scenarios and with various code rates. We also observe the performance of the VFL algorithm with AWGN training noise. Finally, we discuss limitations and future research directions.

*Notation–* $x$ denotes a scalar, $\mathbf{x}$ denotes a vector, and $\mathbf{x}[t]$ denotes the $t$th index of the vector $\mathbf{x}$. The set $\{0, 1, 2, \cdots, N\}$ is denoted by $[N]$. The cardinality of a set $\mathcal{X}$ is denoted $|\mathcal{X}|$. The set $\{x_\ell\}_{\ell=1}^N$ is short hand notation for the set $\{x_1, x_2, \cdots, x_N\}$.

## 2 PROBLEM SETUP

### 2.1 CHANNEL MODEL

We consider the real $L$ user AWGN broadcast channel with noisy feedback (AWGN-BC). That is, there is one transmitter which has $L$ independent, uniformly distributed messages $W_1 \in \mathcal{W}_1$, $W_2 \in \mathcal{W}_2, \cdots, W_L \in \mathcal{W}_L$ that are to be conveyed to receives 1 through $L$, respectively. $\mathcal{W}_\ell$ denotes the set of all messages for receiver $\ell$. At channel use $t \geq 0$, the channel output at receiver $\ell$, $\ell \in [L]$, is given by

$$\mathbf{y}_\ell[t] = \mathbf{x}[t] + \mathbf{n}_\ell^f[t] \tag{1}$$

where $\mathbf{x}[t] \in \mathbb{R}$ is the transmitted symbol at time $t$, $\mathbf{n}_\ell^f[t]$ is i.i.d. noise distributed $\mathbf{n}_\ell^f[t] \sim \mathcal{N}\left(0, \sigma_f^2\right)$ (where the superscript $f$ indicates the noise on the *forward* link). We impose an average transmit power constraint so that

$$\mathbb{E}\left(\sum_{t=1}^N \mathbf{x}^2[t]\right) \leq N \tag{2}$$

where $N$ is the length of the transmission block.

Each receiver has a feedback link to the transmitter that is corrupted with i.i.d. AWGN noise. We assume the channel output given in equation 1 is immediately sent back to the transmitter in a causal

manner. The feedback from receiver $\ell$ is given by

$$\mathbf{z}_\ell[t] = \mathbf{y}_\ell[t-1] + \mathbf{n}_\ell^b[t] \tag{3}$$

where $\mathbf{y}_\ell[t-1]$ is described as in equation 1 and $\mathbf{n}_\ell^b[t]$ is i.i.d. noise distributed $\mathbf{n}_\ell^b[t] \sim \mathcal{N}\left(0, \sigma_b^2\right)$ (where the superscript $b$ indicates the noise on the *backward* link).

## 2.2 CODING DEFINITIONS

We define $R_\ell \in \mathbb{R}^+$ as the rate of transmission for user $\ell$ in bits per channel use. The sum-rate is defined as $R_{sum} = \sum_{j=1}^L R_j$. Then, a $(\lceil 2^{NR_1} \rceil, \lceil 2^{NR_2} \rceil, \cdots, \lceil 2^{NR_L} \rceil, N)$ *code* for the AWGN-BC with feedback consists of

1. A single encoder denoted by functions $f_t(\cdot)$, $t \in [N]$ that maps all the messages $\{W_1, W_2, \cdots, W_L\}$ and the feedback from each user $\{\mathbf{z}_\ell[1], \mathbf{z}_\ell[2], \cdots, \mathbf{z}_\ell[N]\}$, $\ell = 1, \cdots, L$ to $\{\mathbf{x}[1], \cdots, \mathbf{x}[N]\}$ such that the power constraint in equation 2 is obeyed. Specifically, define the encoding procedure at time $t$ as

$$\mathbf{x}[t] = f_t\left(\{W_\ell\}_{\ell=1}^L, \{z_\ell[1], \cdots, z_\ell[t-1]\}_{\ell=1}^L\right)$$

2. $L$ decoders $g_1(\cdot),\ g_2(\cdot), \cdots, g_L(\cdot)$ such that $g_\ell(\cdot)$ maps $\{\mathbf{y}_\ell[1], \cdots \mathbf{y}_\ell[N]\}$ to $\hat{W}_\ell \in \mathcal{W}_\ell$ to decode the message for receiver $\ell$. Specifically, denote the decoding procedure as

$$\hat{W}_\ell = g_\ell\left(\mathbf{y}_\ell[1], \cdots, \mathbf{y}_\ell[N]\right) \tag{4}$$

After decoding, the block error probability for receiver $\ell$ given a message $W_\ell$ is defined as $P_{e,\ell}(W_\ell) := Pr\left(\hat{W}_\ell \neq W_\ell\right)$. In the *global model*, the goal is to design a set of encoding functions $f_t$ for $t \in [N]$ and decoding function $g_\ell$ for each user $\ell$ that minimizes the average probability of error $P_{e,\ell} = \mathbb{E}\left(P_{e,\ell}(W_\ell)\right)$ where the expectation is taken over all possible messages and all users. Specifically, for any encoder-decoder design, we specify that the objective for the broadcast channel is

$$\underset{\{f_t\}_{t \in [N]}, g_1, g_2, \cdots, g_L}{\text{minimize}} \mathbb{E}_\ell\left(P_{e,\ell}\right) \tag{5}$$

$$\text{subject to } \mathbb{E}\left(\sum_{t=1}^N \mathbf{x}^2[t]\right) \leq N$$

# 3 FEEDBACK CODE DESIGN

## 3.1 RPC-BASED BROADCAST CODE (RPC-BC)

In the Robust Power Constrained (RPC) scheme, hereby referred to as RPC-BC, the encoding scheme consist of three layers: 1) a gated recurrent unit layer (GRU), 2) a non-linear layer, and 3) a power control layer. We outline the modules below. Referring to the objective function in equation 5, we see that the optimization problem requires $N$ encoding functions to be designed. To save complexity, a single encoding generation function $f$ is designed such that

$$\mathbf{x}[t] = f\left(\{W_\ell\}_{\ell=1}^L, \{\mathbf{z}_\ell[t-1]\}_{\ell=1}^L, \mathbf{s}[t]\right)$$

where $\mathbf{s}[t]$ is called the *state vector* which propagates over time through a state propagation function $h$ given by

$$\mathbf{s}[t] = h\left(\{W\}_{\ell=1}^L, \{\mathbf{z}_\ell[t-1]\}_{\ell=1}^L, \mathbf{s}[t-1]\right)$$

This state-based encoding procedure is a non-linear extension of the state-space model used for linear encoding in feedback systems, and is intended to capture the time correlation of the feedback signals throughout time Kim et al. (2020); Elia (2004).

1. *GRU layer*: The GRU layer consists of two unidirectional GRUs to capture the time corre-
   lation of the feedback signals causally. The input-output relationship of this layer is given
   by

$$\mathbf{s}_1[t] = \text{GRU}_1\left(\{W_\ell\}_{\ell=1}^L, \{\mathbf{z}_\ell\}_{\ell=1}^L, \mathbf{s}_1[t-1]\right)$$
$$\mathbf{s}_2[t] = \text{GRU}_2\left(\mathbf{s}_1[t], \mathbf{s}_2[t-1]\right)$$

where $\text{GRU}_i$ denotes the function of GRU $i$ and the initial condition for the state vector
and the feedback are given as $\mathbf{s}_i[-1] = \mathbf{0}$ and $\mathbf{z}_i[-1] = \mathbf{0}$. The dimension of the $i$th state
vector is given by $N_{s,i}$. Then, the overall state is represented as

$$\mathbf{s}[t] = [\mathbf{s}_1[t], \mathbf{s}_2[t]] = h\left(\{W_\ell\}_{\ell=1}^L, \{\mathbf{z}_\ell\}_{\ell=1}^L, \mathbf{s}[t-1]\right)$$

where $h$ represents the overall function of the two GRU layers.

2. *Non-linear Layer*: An additional non-linear layer is utilized at the output of the GRUs
   whose input-output relationship is given by

$$\tilde{\mathbf{x}}[t] = \phi\left(\mathbf{w}^T\mathbf{s}_2[t] + b\right)$$

where $\mathbf{w} \in \mathbb{R}^{N_s,2}$ and $b$ are trainable weights and biases, and $\phi$ is a hyperbolic tangent
activation function.

3. *Power Control Layer*: Since power allocation across time is necessary for robust perfor-
   mance Kim et al. (2020), a final power control layer is utilized before transmission. The
   input-output relationship of this layer is given by

$$\mathbf{x}[t] = w_t\gamma_t^{(J)}\left(\tilde{x}[t]\right)$$

where $\gamma_t^{(J)}$ is a normalization function applied to $\tilde{x}[k]$ consisting of sample mean and sam-
ple variance computed from data with size $J$, and $w_t$ is a trainable power weight which
satisfies

$$\sum_{k=1}^N w_t^2 = N$$

It can be shown that with large $J$ the power control layer satisfies the power constraint in
equation 5 almost surely Kim et al. (2020).

The decoding mechanism for each of the $L$ receivers of the RPC-based code consists of 1) a GRU
layer, 2) an attention layer, and 3) a non-linear layer. Each layer is discussed below.

1. *GRU Layer*: Two layers of bi-directional GRUs are used in this layer. The input-output
   relationship of the forward direction of the $\ell$th user are given by

$$\mathbf{r}_{f,1}^\ell[t] = \text{GRU}_{f,1}\left(\mathbf{y}[t], \mathbf{r}_{f,1}^\ell[t-1]\right)$$
$$\mathbf{r}_{f,2}^\ell[t] = \text{GRU}_{f,2}\left(\mathbf{r}_{f,1}^\ell[t], \mathbf{r}_{f,2}^\ell[t-1]\right)$$

and the input-output relationship of the backward direction of the $\ell$th user are given by

$$\mathbf{r}_{b,1}^\ell[t] = \text{GRU}_{b,1}\left(y[t], \mathbf{r}_{b,1}^\ell[t+1]\right)$$
$$\mathbf{r}_{b,2}^\ell[t] = \text{GRU}_{b,2}\left(\mathbf{r}_{b,1}[t], \mathbf{r}_{b,2}^\ell[t+1]\right)$$

where the dimension of $\mathbf{r}_{b,i}^\ell$ and $\mathbf{r}_{f_i}^\ell$ are given by $N_{r,i}^b$ and $N_{r,i}^f$, respectively.

2. *Attention Layer*: The state vectors at the last layer over all communication rounds $t = 1, \cdots, N$ are the input to attention layer. The attention layer is used to capture the long term
   time dependency of the received signals Kim et al. (2020). The input-output relationship
   of this layer for the $\ell$th user is given by

$$\mathbf{r}_{f,att}^\ell = \sum_{t=1}^N \alpha_{f,t}^\ell\mathbf{r}_{f,2}^\ell[t], \quad \mathbf{r}_{b,att}^\ell = \sum_{t=1}^N \alpha_{b,t}^\ell\mathbf{r}_{b,2}^\ell[t]$$

where $\alpha_{f,t}$ and $\alpha_{b,t}$ are trainable attention weights. The final output of this layer is given
by

$$\mathbf{r}_{att}^\ell = [\mathbf{r}_{f,att}^\ell, \mathbf{r}_{b,att}^\ell]^T$$

3. *Non-Linear Layer*: Lastly, a non-linear layer is used to produce the estimate $\hat{W}_\ell$. The input-output relationship of this layer for the $\ell$th user is given by

$$\mathbf{p}_\ell = \theta \left( \mathbf{W}_d^\ell \mathbf{r}_{att}^\ell + \mathbf{v}_d^\ell \right)$$

where $\theta$ is a softmax activation function and $\mathbf{W}_d^\ell \in \mathbb{R}^{|\mathcal{W}_\ell|, N_{r,2}^b + N_{r,2}^f}$ and $\mathbf{v}_d^\ell \in \mathbb{R}^{|\mathcal{W}_\ell|}$ are the trainable weights and biases for user $\ell$. The number of outputs is $|\mathcal{W}_\ell|$ and therefore $\mathbf{p}_\ell$ denotes the probability distribution of the $|\mathcal{W}_\ell|$ possible messages. Fig. 4 in the appendix contains a pictorial representation of the coding scheme.

## 3.2 LIGHTCODE-BASED BROADCAST CODE (LIGHTBC)

In the LightCode based scheme, hereby referred to as LightBC, the encoding scheme consists of two layers: 1) the feature extractor and 2) the MLP, and 3) a power control layer. We outline the modules below.

1. *Feature Extractor*: At each transmission round, a feature extractor (FE) is utilized. The purpose of the FE is to map the data for each message block to a vector representation Ozfatura et al. (2022). The input-output relationship of the FE is given by

$$\mathbf{r}[t] = \text{FE}_e \left( \{W_\ell\}_{\ell=1}^L, \{\mathbf{x}[t-1], \cdots, \mathbf{x}[1]\}, \{\mathbf{z}_\ell[t-1], \cdots, \mathbf{z}_\ell[1]\}_{\ell=1}^L \right)$$

where $\text{FE}_e$ represents the function of the FE at the encoder and the output dimension of $\mathbf{r}[t]$ is $N_{r,e}$. In our simulations, we choose to use the same structure of the FE as in the single user LightCode Ankireddy et al. (2024).

2. *MLP*: After the FE, the signal is fed into a two-layer MLP module whose input-output relationship is given by

$$\tilde{\mathbf{x}}[t] = \text{MLP}_e \left( \mathbf{r}[t] \right)$$

where $\text{MLP}_e$ represents the function of the MLP at the encoder.

3. *Power Control*: Finally, the output $\tilde{\mathbf{x}}[t]$ is fed through a power control layer. We utilize the same power control method as in the RPC coding scheme, given by

$$\mathbf{x}[t] = w_t \gamma_t^{(J)} \left( \tilde{x}[t] \right)$$

where once again $\gamma_t^{(J)}$ is a normalization function applied to $\tilde{x}[k]$ consisting of sample mean and sample variance computed from data with size $J$, and $w_t$ is a trainable power weight which satisfies

$$\sum_{k=1}^N w_t^2 = N$$

The decoding scheme is very similar to that of the encoder, consisting of the same FE module and an MLP. In this case, a single layer MLP is used. Specifically,

1. The FE at the $\ell$th decoder inputs the received symbols across time and outputs a feature vector

$$\mathbf{r}_\ell = \text{FE}_d^\ell \left( \mathbf{y}_\ell[1], \cdots, \mathbf{y}_\ell[N] \right)$$

where $\text{FE}_d^\ell$ represents the function of the FE at the $\ell$th decoder and the output dimension of $\mathbf{r}_\ell$ is $N_{r,d}$.

2. Following the FE, a single layer MLP is used to decode the output. The input output relationship is given by

$$\mathbf{p}_\ell = \text{MLP}_d^\ell \left( \mathbf{r}_k \right)$$

where $\text{MLP}_d^\ell$ represents the function of the MLP at the the $\ell$th decoder. The output of the $\text{MLP}_d^\ell$ module goes through a softmax function, so that the output of the MLP returns a probability vector of length $|\mathcal{W}_\ell|$ of each possible value of $\hat{W}_\ell$.

The decoding and encoding modules are shown pictorially in Figure 5 in the Appendix.

## 4 TRAINING METHODOLOGY

We consider two models for training the codes. The first is a global model, where all parameters are updated according to the same loss function. On the other hand, in practical wireless systems, a federated approach may be more useful and more practical as mentioned in the introduction. Thus, we propose a VFL-like framework in addition to training the global model with uncoded parameter passing between encoder and decoders.

### 4.1 GLOBAL MODEL

In the global model, we consider the objective function in equation 5. Noting that the output of the decoders for each algorithm represent a probability distribution for each of the $|\mathcal{W}_\ell|$ possible message words at the $\ell$th decoder, then the probability of error at the $\ell$th decoder is empirically

$$P_{e,\ell} = \frac{1}{N_{train}} \sum_{x=1}^{N_{train}} \mathbb{1}\left(W_\ell[x] \neq \hat{W}_\ell[x]\right)$$

where $\mathbb{1}(\cdot)$ is an indicator function that is 1 when the argument is true, zero otherwise, and $W_\ell[x]$ is the true message vector for sample $x$ and $\hat{W}_\ell[x]$ is the decoded message vector for sample $x$. The overall expected probability of error over all users is then given by

$$\mathbb{E}_\ell\left(P_{e,\ell}\right) = \frac{1}{L} \sum_{\ell=1}^{L} P_{e,\ell}$$

Since each decoder is essentially performing its own classification problem, then, like the single user case, we can define the individual loss function of the $\ell$th user using the cross-entropy loss as

$$L_{CE}^\ell = \frac{1}{N_{batch}} \sum_{x=1}^{N_{batch}} \left(\sum_{n=1}^{|\mathcal{W}_\ell|} c_{xn}^\ell \log p_{xn}^\ell\right)$$

where $N_{batch}$ is the batch size, $c_{xn}$ is the actual probability of message vector is $x$ at sample time $n$, $P(W_\ell[n] = x)$, and $p_{xn}$ is the predicted probability that $W_\ell[n] = x$ for sample $n$. Treating all users as equally important, the global loss is

$$L_{CE} = \frac{1}{L} \sum_{\ell=1}^{L} L_{CE}^\ell \tag{6}$$

By letting $c_{xn}^\ell = 1$ only for $W_\ell[n] = x$, else zero, the objective of minimizing the probability of error in equation 5 is instead transformed into a classification problem.

### 4.2 FEDERATED MODEL

We also train the RPC-BC model using a vertical federated learning approach. Assume that the $\ell$th decoder has its own local model $\mathcal{G}_\ell$ parameterized by $\theta_\ell$, and the encoder has its own model $\mathcal{F}$ parameterized by $\theta_e$. We argue that since each decoder is attempting to minimize its own probability of error $P_{e,\ell}$, it is not necessary for each decoder $\ell$ to store a global model. On the other hand, since the encoder is sending one signal to all decoders, the encoder should own a global model in order to contribute to minimizing the overall probability of error for all decoders.

Let $N_{batch}$ be the communication batch size. Then, over $N_{batch} \times N$ communication rounds, the encoder broadcasts $N_{batch}$ codewords to the $L$ users, where each codeword is transmitted over $N$ channel uses. We assume that each of the $\ell$ users does not know the intended codeword *a priori*. We shall call each set of $N$ channel uses a *sample*. For the $n$th sample of the $N_{batch}$ samples, each local model $\mathcal{G}_\ell$ computes its output $\mathcal{H}_\ell[n] = \mathcal{G}_\ell\left(W_\ell[n], \theta_\ell\right)$. This output $\mathcal{H}_\ell[n]$ is transmitted via the feedback link to the encoder. After the $N_{batch} \times N$ communication rounds, and with all of the outputs from each decoder, the encoder computes the overall loss of the system according to equation 6. The encoder computes the gradients of its global model $\mathcal{F}$ and then updates its model.

| LightBC | | RPC-BC | |
| --- | --- | --- | --- |
| Batch Size | $10^5$ | Batch Size | $10^5$ |
| Total Epochs | 120 | Total Epochs | 100 |
| $N_{train}$ | $10^8$ | $N_{train}$ | $10^7$ |
| Learning Rate | $10^{-3}$ | Learning Rate | $10^{-2}$ |
| Optimizer | AdamW | Optimizer | Adam |
| Scheduler | LambdaLR | Scheduler | StepLR |

Table 1: Training Parameters for RPC-BC and LightBC

Then, the encoder computes the loss with respect to each local model $\mathcal{G}_\ell$ and computes the gradients with respect to each local model. These gradients are transmitted to each receiver, which then update their model $\mathcal{G}_\ell$ accordingly. The training process is shown in Figure 6 in the appendix.

Note that in this process, noise may impact both the value of the model output $\mathcal{H}_\ell$ which is sent to the encoder, as well as the gradients which are transmitted to each decoder. We note that the output of the decoder is a $|W_\ell|-$length vector of probabilities. The decoder sends the length $|W_\ell|-$length vector of probabilities over $|W_\ell|-$ channel uses. Likewise, when the gradients are computed for each decoder, they must also be transmitted to each respective decoder. We assume that the gradients are sent across the downlink in an uncoded manner in an orthogonal fashion, such as time division duplexing (TDD). In the federated approach, we assume that the batch size $N_{batch}$ is the same as the global model, outlined in the training parameters table. When sending the gradients from the encoder to the decoders, we scale the gradients to obey an average power constraint. That is, if the gradient vector for the $\ell$th user are given by $\theta'_\ell = \frac{\partial L^\ell_{CE}}{\partial \theta_\ell}$, then the transmission power $P_{grad}$ is scaled such that

$$P_{grad}\|\theta'_\ell\|_2^2 = N_{\ell,grad} \tag{7}$$

somewhat like the power constraint in equation 2, where $N_{\ell,grad}$ is the length of the gradient vector for the $\ell$th user.

The details of the training parameters for RPC-BC and LightBC are outlined in Tables 1 and 2. We keep the training parameters relatively consistent with the training parameters proposed in the single user versions of these codes. The outline for training each model is given in the Appendix.

## 5 NUMERICAL EXPERIMENTS AND DISCUSSION

In this section, we simulate the performance of Light-BC and RPC-BC for various code rates and noise regimes. We assume that in the broadcast case, there are 2 receivers, though we note that either code may be extended to an arbitrary number of users. In the following, we refer to the number of message bits as $K$, given by $K = \log_2(|W_\ell|)$. In our training, we set $N_{inference}$ to $10^{-8}$ with the rest of the training parameters outlined in Table 1. In the inference stage, the $\ell$th decoder chooses the message corresponding to the entry of $\mathbf{p}_\ell$ with the highest value as $\hat{W}_\ell$. In some cases, we compare results against the concatenated scheme based off linear codes, using the closed form SNR expressions for the concatenated coding scheme of the symmetric AWGN-BC with noisy feedback for the scheme when $\lambda \to 0$ and $\tilde{L} \to \infty$ (see (42) in Ahmad et al. (2015)). In the linear scheme, we assume that the signal is modulated to a $2^K-$PAM symbol and transmitted using the concatenated scheme outlined by equation 41 in Ahmad et al. (2015).

### 5.1 PERFORMANCE WITH NOISELESS FEEDBACK

First, we compare the performance of LightBC versus RPC-BC in the noiseless feedback case for sum rates $R = 1/3$ and $R = 2/3$. For the sum rate $R = 1/3$ case, we let $K = 3$ per user and set $N = 18$, while for the sum rate $R = 2/3$ case, we let $K = 6$ per user and keep $N = 18$. The forward SNR is swept from $-3$ to $1$ dB. Fig. 1 shows the results of the experiments. We see for the low rate case of 2 dB, at low SNR, the RPC-BC code performs better, but from $-1$dB and beyond, no errors occurred in the inference stage (that is, the probability of error is less than $10^{-8}$). For this lower SNR region, both RPC-BC and LightBC outperform the concatenated scheme, but

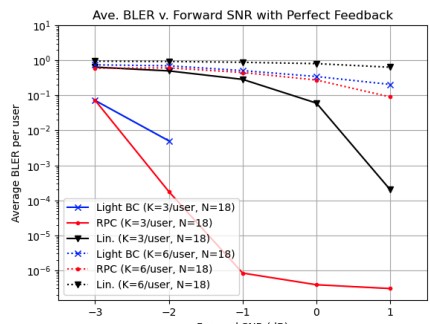

Figure 1: LightBC versus RPC-BC with Perfect Feedback

rate $R = 2/3$ appears to be too high of a rate code for this SNR region in all cases. We note that LightBC behaves somewhat like a linear feedback code in that its performance is more severely impacted by noise as opposed to RPC-BC, but its performance rapidly improves as the channel gets more reliable.

## 5.2 PERFORMANCE WITH NOISY FEEDBACK

Here, we compare the performance of LightBC versus RPC-BC in the noisy feedback case for sum rates $R = 1/3$, , $R = 5/9$, and $R = 2/3$. We also compare the broadcast codes against TDD for rate $2/3$, meaning that instead of using a broadcast code for two users for $N = 18$ communication rounds, the transmitter sends to each of two users over 9 communication rounds using a non-broadcast code. It is seen in Fig. 2 that for the rate $R = 2/3$ RPC-BC code, the RPC-BC code outperforms TDD. On the other hand, in the rate $R = 2/3$ regime, LightBC performs around the same or slightly worse than the single user LightCode.

In the rate $R = 1/3$ regime, we see in Fig. 2, that LightBC outperforms RBC-BC as the feedback noise becomes smaller, where at $-20$dB noise power and smaller, the average BLER falls below $10^{-8}$. For the rate $R = 5/9$ regime, we see that for low feedback noise, LightBC outperforms RPC-BC, but as the feedback noise increases, the RPC-BC slightly outperforms LightBC. In most cases, both RPC-BC and LightBC outperform the concatenated scheme, except in the low rate regime of $R = 1/3$, where as the feedback noise tends to 0, the probability of block error also tends to 0. Once again, LightBC demonstrates linear-like code behavior, where the probability of error drastically improves with improving channel conditions. For example, there is a steep performance improvement past a certain feedback threshhold for all of the rates in Fig. 2 (b).

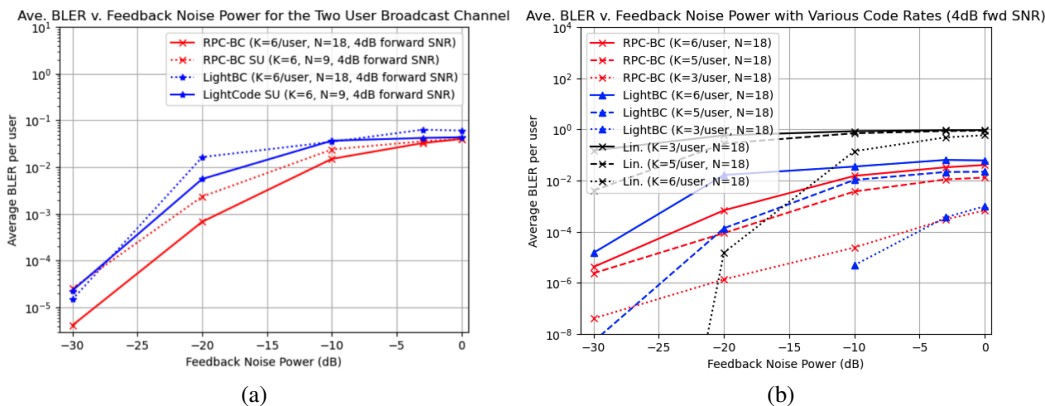

(a)    (b)

Figure 2: (a) Comparison of Performance with Noisy Feedback for Rate $R = 2/3$ Against TDD Scheme and (b) Comparison of Performance with Noisy Feedback for Different Code Rates

## 5.3 FEDERATED LEARNING APPROACH

Now, we train the RPC-BC and the LightBC code using the proposed federated approach. We show the performance of both using rate $R = 2/3$ and scale the power accordingly using equation 7 to simulate the desired SNR when transmitting the gradients. It can be seen in Fig. 3, the performance is considerably degraded with noise in the uncoded transmission of parameters as the average BLER increases by orders of magnitude. When the SNR when transmitting gradients is high, LightBC performs well relative to the global baseline. However, it is not a realistic assumption that the downlink will be extremely reliable in practice. In both cases, it seems that the models are sensitive to training noise and more reliable methods for passing model parameters between users needs to be developed for training broadcast codes in practice.

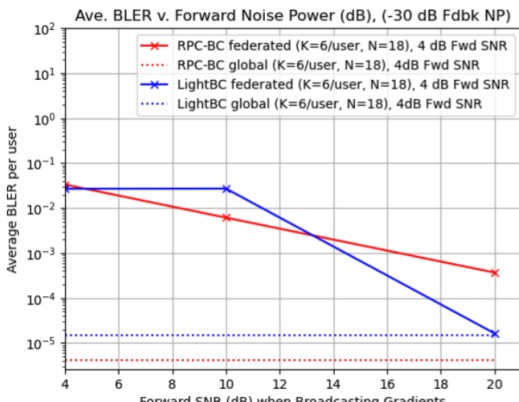

Figure 3: Performance with Federated Training Scheme and Uncoded Transmission of Training Parameters

## 6 CONCLUSION

In this work, we have extended deep-learning aided feedback codes to broadcast channels and evaluated their performance. Our numerical studies indicated that there appears to be not much advantage over using the simple extension of LightBC in the broadcast setting versus utilizing TDD with the single user LightCode in the high rate, noisy regime. On the other hand, with certain noise regimes and code rates, there does appear to be an advantage in using RPC-BC versus TDMA with the single user RPC. The experiments indicate that in especially noisy environments and higher code rates, RPC-BC tends to outperm LightBC, whereas LightBC tends to perform exceptionally well in lower noise, lower code rate scenarios. The more robust performance in higher noise scenarios of RPC-BC could possible be attributed to the RNN architecture of RPC-BC since it allows noise averaging across communication rounds. On the other hand, LightBC behaves much like linear feedback codes in that its probability of error steeply drops off as the channel gets more reliable. Nonetheless, this seems to suggest that more work needs to be done tailoring Deep Learning algorithms specifically to the broadcast communication scheme in order to improve performance in more unreliable communication settings, where it may be necessary to employ RNN-based models in especially unreliable channels to leverage noise averaging.

In addition, we also explored the performance of a federated approach to training each of the codes where AWGN noise was added to parameters when being passed between encoder and decoders. We found that the addition of AWGN to the gradients and feedback channel generally resulted in considerable performance degradation, suggesting that a reliable communication protocol of model parameters is necessary when using federated learning. Though a federated approach makes sense for training deep learned codes, one practical restriction is that decoders that may leverage noise averaging such as in RPC-BC typically have many parameters due to the hidden states in the GRUs. Thus, more research is necessary to compress parameters or design lower complexity decoder modules that perform well in high noise scenarios.

REPRODUCIBILITY STATEMENT

The code for both the global and federated approach to LightBC and RPC-BC are included in the supplementary materials via a link to an anonymous repository which may also be accessed at https://anonymous.4open.science/r/ICLRsub-D721/.

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

## A    DIAGRAMS OF RPC-BC AND LIGHTBC

Here we include the diagrams for the RPC-BC and LightBC architectures, respectively. In Fig. 4, $N_{hidden}$ refers to the dimension of the hidden state in each GRU module.

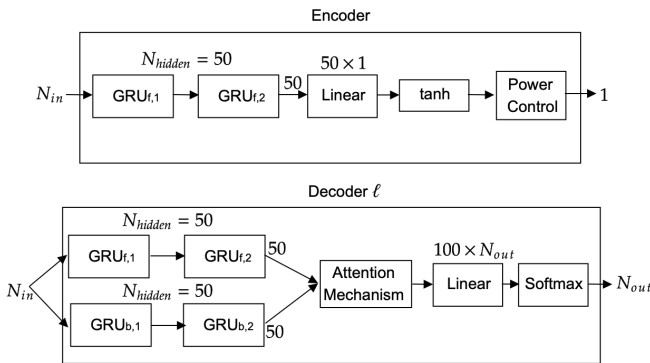

Figure 4: Encoder and Decoder Diagram for the RPC-BC scheme

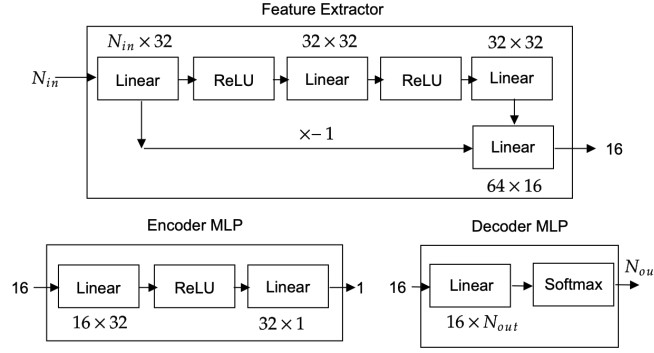

Figure 5: Feature Extractor Module and Encoder/Decoder MLP diagrams for LightBC

## B    TRAINING RPC-BC AND LIGHTBC

Here, we give the outline for the training process for LightBC and RPC-BC trained against the global model in Algorithm 1. Figure 6 gives a pictorial representation of the federated training process, and the details of the federated training process are outlined in Algorithm 2.

Figure 6: Federated Training Process Diagram

---

**Algorithm 1** Training RPC-BC and LightBC: Global

---

1: **Input:** Encoder Model, Decoder Models, $K$ bits per user, $L$ users, noise variances $\sigma_b^2$, $\sigma_f^2$, training parameters, number of epochs $E$, batch size $N_{batch}$, number of training samples $N_{train}$
2: **for** $e \leq E$ **do**
3:     **for** $n \leq N_{batch}/N_{train}$ **do**
4:         Generate $N_{batch}$ random messages for each of $L$ users
5:         **for** $t \leq N$ **do**            $\triangleright$ Iteratively code across communication rounds
6:             $\mathbf{x}[t] = f(\{W_\ell\}_{\ell=1}^L, \{\mathbf{z}_\ell[t-1]\}_{\ell=1}^L, \mathbf{s}[t])$     $\triangleright$ Encode during channel use $i$
7:             $\mathbf{y}_\ell[t] = \mathbf{x}[t] + \mathbf{n}_\ell^f[t]$
8:             $\mathbf{z}_\ell[t] = \mathbf{y}_\ell[t-1] + \mathbf{n}_\ell^b[t]$
9:         **end for**
10:         $\mathbf{p}_\ell = g_\ell(\mathbf{y}_\ell[1], \cdots, \mathbf{y}_\ell[N])$             $\triangleright$ Decode after all rounds
11:         Compute the cross entropy loss $L_{CE} = \frac{1}{L}\sum_{\ell=1}^L L_{CE}^\ell$
12:         Clip gradients (.5 for LightBC, 1 for RPC-BC)
13:         Update parameters for Encoder $f$ and Decoders $g_\ell$ with specified optimizer and learning rate.
14:     **end for**
15:     Update learning rate with specified scheduler.
16: **end for**

---

---

**Algorithm 2** Training RPC-BC: Federated

---

1: **Input:** Encoder Model, Decoder Models, $K$ bits per user, $L$ users, noise variances $\sigma_b^2$, $\sigma_f^2$, training parameters, number of epochs $E$, batch size $N_{batch}$, number of training samples $N_{train}$
2: **for** $e \leq E$ **do**
3:     **for** $n \leq N_{batch}/N_{train}$ **do**
4:         Generate $N_{batch}$ random messages for each of $L$ users
5:         **for** $t \leq N$ **do**                       $\triangleright$ Code across communication rounds
6:             $\mathbf{x}[t] = f(\{W_\ell\}_{\ell=1}^L, \{\mathbf{z}_\ell[t-1]\}_{\ell=1}^L, \mathbf{s}[t])$       $\triangleright$ Encode during channel use $i$
7:             $\mathbf{y}_\ell[t] = \mathbf{x}[t] + \mathbf{n}_\ell^f[t]$
8:             $\mathbf{z}_\ell[t] = \mathbf{y}_\ell[t-1] + \mathbf{n}_\ell^b[t]$
9:         **end for**
10:         $\mathbf{p}_\ell = g_\ell\left(\mathbf{y}_\ell[1], \cdots, \mathbf{y}_\ell[N]\right)$                 $\triangleright$ Decode after all rounds
11:         $\hat{\mathbf{p}}_\ell = \mathbf{p}_\ell[t'] + \mathbf{n}_\ell^b[t'], \; t' = 0, \cdots, |\mathcal{W}_\ell| - 1$     $\triangleright$ Send decoded output to encoder
12:         Compute the cross entropy loss at encoder $L_{CE} = \frac{1}{L}\sum_{\ell=1}^L L_{CE}^\ell$
13:         Compute decoder gradients $\theta_\ell' = \frac{\partial L_{CE}^\ell}{\partial \theta_\ell}$
14:         Clip gradients (.5 for LightBC, 1 for RPC-BC)
15:         Transmit decoder gradients $\hat{\theta}_\ell' = \theta_\ell + \mathbf{n}_\ell^f[t'], t' = 0, 1, \cdots, N_{\ell,grad} - 1$ (with appropriate power scaling to achieve desired SNR)
16:         Update parameters for Encoder $f$ and Decoders $g_\ell$ with specified optimizer and learning rate.
17:     **end for**
18:     Update learning rate with specified scheduler.
19: **end for**

---

