# OpenReview forum: "Deep Learning Aided Broadcast Codes With Feedback"
_ICLR.cc/2025/Conference — ICLR 2025 Conference Withdrawn Submission_

### Official Review · Reviewer_qLek · 2024-10-30

**Soundness:** 3
**Presentation:** 2
**Contribution:** 2
**Rating:** 5
**Confidence:** 5

**Summary:**

The authors propose a new feedback coding scheme for AWGN broadcast channels (AWGN-BC) by extending two of the exising feedback coding schemes for feedback channels, a RNN based scheme and a lightweight-MLP scheme. Further, a new vertical fedearted learning based training scheme is proposed, which reflects a real world scenario more closely. Finally the authors peroform an emoirial study to demonstrate the benefits of each coding scheme under different channel conditions.

**Strengths:**

1. Relatively unexplored problem space. While deep learning based feedback schemes are well studied for single user AWGN channels, utility of the same for broadcast channels is not well studied and open for improvements.
2. The paper is well written overall and easy to follow for someone familiar with the problem space.
3. The concept of federated training of encoder and decoders is novel and interesting and relevant to practical settings.
4. Performance gains over existing linear schemes is non-trivial and compelling.
5. The takeway on LightBC being a better choice in low-noise scenarios and RPC-BC being better in high-noise scenarios is sn interesting observation.

**Weaknesses:**

1. Limited novelty from an architecture and training perspective, as both RPC and LightCode schemes are being reused directly from existing works.
2.  No system model included in the main body of the paper, making the problem setup hard to grasp.
3. Throughout the paper, there was the mention of LightCode being simpler than RPC. But there is not study on the complaexity, memory, and run-time comparison between the schemes makaing it harder to understand the difference between the schemes in terms of resource overhead.
4. Authors mention in introduction that the capacity of AWGN-BC channel increases in presence of feedback. In the experiemnts section, it would be interesting to see some discussion on this and how the feedback is impacting the performance. Specifically, a reference line indicating the capacity of the channel would make it easier to comprehened the efficiency of the proposed scheme compared to the theoretical limits.
5. Captions to the figures are rather short and very uninformative. I suggest updating all the captions to reflect the key takeaway.
6. Experimental section is written poorly. It's not immediately apparent what authors mean by "Lin." in the plots. More discussion should go into explaining the figures well.
7. Lack of proper baselines. Since the authors propose modifying two schemes that were originally proposed for single user case, there should be atleast one more learning-based baseline that was designed for 2-user case. The reference Li. et al. "Deep learning-aided coding for the fading broadcast channel with feedback" would be a good baseline (despite the BPSK modulation).

Misc.
Minor typos and grammatical errors should be corrected throughout the paper. ex. in line 147 - "receives"

**Questions:**

Included everything in the weakness section

---

### Official Review · Reviewer_mGbG · 2024-11-02

**Soundness:** 2
**Presentation:** 1
**Contribution:** 1
**Rating:** 3
**Confidence:** 5

**Summary:**

This paper studies error correction coding over a broadcast channel in the presence of feedback. The goal of the transmitter is to convey independent messages to different receivers using the same channel resources. Assuming that the transmitter can see the channel outputs received by the receivers perfectly or with some noise, its goal is to transmit signals in an iterative fashion to gradually improve the decoding reliability of the receivers of their own messages. The author review the literature in this domain, and emphasize the lack of prior work on deep learning aided code design for this problem, although some works have appeared for the single-user channel with feedback.

The authors extend two approaches recently proposed for the single-user problem to the broadcast channel setting. They also propose a federated learning based training approach for the code.

**Strengths:**

The main strength of the paper is to study a communication scenario that has not received much attention, particularly in terms of the application of recently developed neural network based code designs.

**Weaknesses:**

- The novelty of the paper is very limited. The authors mainly take two existing designs and train these models by considering multiple receivers. This is rather trivial, and as such the paper does not introduce any new concept, architecture or tool.

- The federated learning (FL) approach is not well motivated or explained. Why would a vertical FL approach make sense here? As long as the channel model is available, what prevents the encoder from training all the decoders centrally? Is it a complexity issue? The proposed uncoded transmission of gradients, as also observed by the authors, is limiting, and not well motivated.

- Presentation can be improved. Especially in the numerical results part, there are some confusing sentences.

- Comparison is limited to a single relatively weak code from Li et al. It seems that the state of the art for point-to-point channels with feedback (GBAF codes) is not considered in this scenario. Although complexity is argued against these codes, there is no presentation of complexity for the presented schemes.

**Questions:**

- Can you please better motivate the FL approach? Is the scenario a distributed communication setting with unknown channels? I believe estimating the channels before training the code would be a better approach in general (as is done in practice). Can you please better motivate the underlying scenario?

- RPC-BC seems to do worse than time-division transmission of point-to-point RPC codes. Then, what is the point of considering these codes? LightBC seems to improve compared to time-division, does that hold for longer codes as well? I expect the training will get harder quickly with the code length, and the gains with respect to TD may quickly diminish. Can you please provide additional simulation for longer code lengths?

- Why consider the code rate of 2/3 in Fig. 1 given that the error probabilities are quite high?

- Can you please provide a comparison of these schemes with the linear coding approaches proposed in the literature? Even though you argue that non-linear coding can help, you do not provide any evidence for that.

- How practical these codes in practice? If I understand correctly, a different neural network decoder is trained for each receiver. In a practical setting, where users roam from cell to cell, does it mean that each user has to have all the decoders as they can be assigned as any user i. Similarly, it seems that a separate encoder/decoder are trained for each SNR, which further increases the memory requirement.

---

### Official Review · Reviewer_z6Kk · 2024-11-05

**Soundness:** 2
**Presentation:** 3
**Contribution:** 2
**Rating:** 3
**Confidence:** 3

**Summary:**

The paper considers a broadcast channel with feedback and considers a two-user symmetrical AWGN case. Feedback is known to improve the capacity region of this channel, and the authors consider a machine learning-based approach to construct good codes for this setup. The authors consider both noiseless and noisy feedback.

The proposed training methods are based on the extension of previous works devoted to a single-user channel:
 - Robust Power-Constrained Deep Learning Algorithm (RPC)
 - LightCode

The authors consider a simulation setup with small block length equal to 9 or 18 channel uses. The authors consider three different scenarios:
- noiseless feedback,
- noisy feedback and comparison with an orthogonal (TDD) scheme,
- noisy feedback and comparison between different coding rates.

**Strengths:**

The authors consider the case of a multiuser broadcast channel with feedback. Feedback is known to improve the capacity region of the multiuser channel without feedback. Moreover, the behavior of this setup in the finite blocklength regime is not addressed in the literature. Thus, the application of machine learning in designing codes for broadcast channels with feedback seems quite reasonable and interesting.

**Weaknesses:**

The paper lacks some discussion related to the broadcast channel with feedback. The authors just say that «the use of feedback in the AWGN-BC channel can far exceed the capacity of the AWGN-BC channel without feedback». I think that more discussion on this topic may significantly improve the understanding of the problem and numerical results analysis. The questions that I suggest to address may be as follows:
- What are existing theoretical results on capacity region?
- How exactly the capacity region can be improved and in which setups?
- What about successive interference cancellation that is widely used in communications? Is it applicable to feedback channels?
- What about finite block length regime? Were there any attempts to address it?

Next, in the final section, the authors mention that "... numerical studies indicated that there appears to be not much advantage over using the simple extension of LightBC in the broadcast setting versus utilizing TDD with the single user LightCode in the high rate, noisy regime", which seems a bit confusing. It seems that the improvement can be achieved exactly when a non-orthogonal transmission scheme is used.

The authors consider numerical results by addressing a finite (N = 18 channel uses) block length, which is too far from any coding scheme used in practice. I suggest adding some discussion on how the block length can be increased and which results one should expect in this case.

The main numerical results are presented in Fig. 1 (noiseless feedback), and this figure lacks a comparison with any known theoretical bounds (like orthogonal TDM mentioned later when considering noisy feedback channels or a finite block length random coding bound for multiple access channels without feedback) and practical schemes. More references will also improve the understanding of the provided results.

**Questions:**

1. Can the scheme be applied to a block transmission with feedback? I think this scenario is of high importance because cellular systems operate in this regime (hybrid ARQ).
2. Please provide a more detailed comparison with orthogonal (TDM) schemes when considering noiseless feedback channel. Comparing results of the  LightCode paper (arXiv 2403.10751) and Fig. 5 (rate K=3/N=9, and BLER=1e-9 at approximately -1 dB SNR), and results presented in Fig 1. (BLER 1e-6 at the same SNR for K=3/uers and N=18 channel uses).
3. Please address the problem of error-floor appearing in Figure 1. (RPC, K=3/N=18).

---

### Official Review · Reviewer_kBDJ · 2024-11-06

**Soundness:** 1
**Presentation:** 1
**Contribution:** 1
**Rating:** 3
**Confidence:** 3

**Summary:**

This manuscript proposes two classes of deep-learning assisted encoder and decoders for a broadcast channel. The first is an extension of the RNN-based architecture, and the second is a lighter MLP-based architecture. The authors compare them with conventional concatenated codes in various settings, indicating that the proposed architectures outperform conventional codes. In addition, they consider the federated learning scheme for the models and examine the performance when the gradient of trainable weights is transmitted over noisy channels.

**Strengths:**

* The proposed models are the first fully-learnable architectures for an AWGN-broadcast channel (AWGN-BC). They show excellent performance compared with a conventional concatenated linear code.
* The authors investigate a federated learning strategy for training weights in the models. In particular, they examine the effect of noise in transmitting gradients in the training process.
* These issues are of importance in terms of future wireless communications.

**Weaknesses:**

Although the reviewer agrees with the importance of the issues, there are several flaws in this manuscript.
1. **Poor technical contributions**

The contributions of this manuscript are twofold: the proposal of deep-learning architectures for AWGN-BC and the proposal of federated learning for these models. However, the proposed architectures are simple extensions of existing models for single-user cases. In short, the modifications are only learning multiple decoders and changing the loss function. As for federated learning, the concept is familiar and the modifications of the learning process are straightforward. Including not providing any theoretical analyses, the reviewer cannot avoid judging that the contributions of this manuscript are poor.

2. **Lack of comparison between proposed models**

The authors proposed two models for AWGN-BC. It is claimed that LightBC is lighter than RPC-BC, but the authors do not explicitly compare the number of trainable weights and/or training costs of these models. This will be important because the comparison of model size is a performance metric other than an error rate. In addition, it will be related to the analysis of the numerical results, e.g., Fig. 3.

3. **Lack of comparison with other models**

Another issue is that the numerical results do not contain those of other learning-based AWGN-BC codes such as [Li et al. 2022] in the manuscript. Even if Li's model is not fully learnable, as stated in Sec. 1.1, there is no reason to omit the model from comparison in this manuscript.

4. **Possibly insufficient numerical experiments on federated learning**

The authors state, "We choose to explore the uncoded downlink versus a quantized method as it has been shown that there is better convergence behavior in federated learning with noisy downlinks Amiri et al. (2021)." in Line 126-129. However, Fig. 3 only contains the results of uncoded transmission (Line 490). The manuscript seems to be flawed.

5. **No theoretical analysis nor interpretation of numerical results**

The manuscript contains no theoretical analysis of the proposed models or federated learning process. At the very least, I believe that the interpretation of the numerical results should be described in the manuscript. However, the manuscript only describes the facts from numerical results. See the "Question" section for more details.

**Questions:**

Questions:
1. In some numerical results, RPC-BC shows an error floor in the high SNR region but LightBC does not, which is described by "LightBC behaves somewhat like a linear feedback code" (Line448). The reviewer wonders the reason of the fact because RPC-BC must be more flexible than LightBC. A possible reason is the hardness of learning RPC-BC. Is there any explanation of the fact?
Another question: why LightBC can behaves like linear codes? Does learned LightBC really have some linearlity like a linear code? How to examine it?

2. Is there any possible reason that LightBC outperforms RPC-BC in some settings? Is it related to the hardness of learning models?

3. In Fig.3, why LightBC by federated learning performs close to LightBC by global learning?

4. What is the benefit of fully-learnanable AWGN-BC code?

5. Are the training parameters in Tab. 1 optimized for AWGN-BC? The reviewer is concerned that the use of parameters "consistent with the training parameters proposed in the single user versions" (Line 408) may be unsuitable for the AWGN-BC case, resulting in poor performance of RPC-BC.

Suggestions:
1. The size of trainable parameters should be described. It will be helpful to show how much LightBC is lighter than RPC-BC.
2. In Line 232, $\mathbb{R}^{N_s,2}$ should be $\mathbb{R}^{N_{s,2}}$.
3. In Line 394, "The decoder sends the length $|W_\ell|$−length vector..." is somewhat confusing.
4. In Fig. 2(a), the term "SU" is confusing because it is denoted by "TDD" in the main text.

---

### Note · Authors · 2024-11-24

I have read and agree with the venue's withdrawal policy on behalf of myself and my co-authors.